# Identification of Kukoamine A, Zeaxanthin, and Clexane as New Furin Inhibitors

**DOI:** 10.3390/ijms23052796

**Published:** 2022-03-03

**Authors:** David Zaragoza-Huesca, Carlos Martínez-Cortés, Antonio Jesús Banegas-Luna, Alfonso Pérez-Garrido, Josefina María Vegara-Meseguer, Julia Peñas-Martínez, Maria Carmen Rodenas, Salvador Espín, Horacio Pérez-Sánchez, Irene Martínez-Martínez

**Affiliations:** 1Servicio de Hematología y Oncología Médica, Hospital Universitario Morales Meseguer, Centro Regional de Hemodonación, CIBERER, Universidad de Murcia, IMIB-Arrixaca, 30003 Murcia, Spain; davidzaragozahuesca5369@gmail.com (D.Z.-H.); julia.penas@um.es (J.P.-M.); mariacarmen.rodenas1@um.es (M.C.R.); salvaalmudema@gmail.com (S.E.); 2Structural Bioinformatics and High Performance Computing Research Group (BIO-HPC), Computer Engineering Department, UCAM Universidad Católica de Murcia, 30107 Guadalupe, Spain; cmartinez1@ucam.edu (C.M.-C.); ajbanegas@ucam.edu (A.J.B.-L.); aperez@ucam.edu (A.P.-G.); jvegara@ucam.edu (J.M.V.-M.)

**Keywords:** furin, virtual screening, inhibitors, CMK

## Abstract

The endogenous protease furin is a key protein in many different diseases, such as cancer and infections. For this reason, a wide range of studies has focused on targeting furin from a therapeutic point of view. Our main objective consisted of identifying new compounds that could enlarge the furin inhibitor arsenal; secondarily, we assayed their adjuvant effect in combination with a known furin inhibitor, CMK, which avoids the SARS-CoV-2 S protein cleavage by means of that inhibition. Virtual screening was carried out to identify potential furin inhibitors. The inhibition of physiological and purified recombinant furin by screening selected compounds, Clexane, and these drugs in combination with CMK was assayed in fluorogenic tests by using a specific furin substrate. The effects of the selected inhibitors from virtual screening on cell viability (293T HEK cell line) were assayed by means of flow cytometry. Through virtual screening, Zeaxanthin and Kukoamine A were selected as the main potential furin inhibitors. In fluorogenic assays, these two compounds and Clexane inhibited both physiological and recombinant furin in a dose-dependent way. In addition, these compounds increased physiological furin inhibition by CMK, showing an adjuvant effect. In conclusion, we identified Kukoamine A, Zeaxanthin, and Clexane as new furin inhibitors. In addition, these drugs were able to increase furin inhibition by CMK, so they could also increase its efficiency when avoiding S protein proteolysis, which is essential for SARS-CoV-2 cell infection.

## 1. Introduction

Furin is a proprotein convertase whose main function consists of cleaving zymogens that, after cleavage, acquire biological active function [1]. The furin cleavage site in substrates presents arginine at the first and fourth amino acids from the N-terminal region [2]. Among these substrates, we highlight growth factors, cytokines, coagulation proteins, albumin, hormones, metalloproteinases, and many types of receptors [3]. Since furin acts on a wide variety of substrates, it regulates many different processes. For example, furin regulates pancreatic granular acidification and, consequently, it also affects insulin secretion [4]. This enzyme is also implicated in proliferation and apoptosis processes, as in granulosa cells from ovaries in rats [5]. Even during embryonic development, furin plays an important role, since it increases the migration and expansion of human trophoblast cells [6].

Apart from its physiological relevance, furin is also a key molecule in cancer development and infectious diseases. In the case of cancer, it promotes carcinogenesis and invasion in many types of tumors [7]. For example, in non-small-cell lung cancer, furin expression is correlated with tumor invasion [8]. In head and neck squamous cell carcinomas, furin promotes tumor proliferation and invasiveness [9,10,11]. In colon cancer, this enzyme is important for tumor growth, metastasis, and angiogenesis [12,13]. With respect to infectious diseases, many pathogen molecules can be activated when cleaved by furin, acquiring the conformation they need to promote infection [1]. This has been related to many types of bacterial toxins, such as AB toxins [1], which need to be cleaved to become active [14]. Other examples are diphtheria and *Pseudomonas* toxins [15,16]. In the case of viruses, glycoproteins located at the envelope are also processed by furin [1], as in infections caused by herpesvirus [17], coronavirus [18], or bornavirus [19]. Specifically, S protein from severe acute respiratory syndrome coronavirus 2 (SARS-CoV-2) is another protein activated by furin [2]. In the infectious process, when S protein joins to its cellular receptor, angiotensin-converting enzyme 2 (ACE2), it exposes the binding site between S1 and S2 domains, which is then accessible to proteases such as furin. When that binding site is cleaved, S2 acquires the capacity to initiate membrane fusion between the host cell and virus, which injects its genetic material into the cytoplasm [20]. Based on these data, furin is considered as an enzyme implicated in SARS-CoV-2 infection and its consequent disease called COVID-19, which has led many countries to their sanitary capacity limits, even going so far as to misadjust the preventive system in many nations [21]. This pandemic spread very rapidly throughout the world. Since its emergence in December 2019, 16.5 million people were infected in just 8 months, seriously affecting the elderly population and health professionals who were in contact with SARS-CoV-2 patients [22]. Due to these data and the lack of vaccines during the first months after the pandemic started, different known biomolecules were used as potential anti-COVID-19 agents, such as herbs from *Chrysanthemi flos*, *Erigeron breviscapus,* or *Coptidis rhizome* [23]. To date, the World Health Organization has reported more than 5.5 million deaths globally [24], and the demonstrated capacity of SARS-CoV-2 to generate new variants keeps health organizations on alert since different mutations from these variants can affect vaccine efficacy [25]. 

Considering all these data concerning furin implication in different diseases, especially in the recent pandemic called COVID-19, different drugs have been studied to inhibit it, such as diminazene, which is currently used as an anti-parasitic drug, Decanoyl-Arg-Val-Lys-Arg-chloromethylketone (CMK), modified α1-antitrypsin Portland [2] or low molecular weight heparin (LMWH) [26]. Diminazene was discovered as a furin inhibitor by virtual screening and validated in an inhibition assay against purified furin [2]. CMK has already been tested as a furin inhibitor that, in addition, is capable of avoiding SARS-CoV-2 S protein cleavage in a cellular model with a 50 µM concentration [27]. The drug α1-antitrypsin Portland inhibits furin by slow and strong binding, and it exerts a suicide inhibitory mechanism as do other serine protease inhibitors [28]. A study on heparin supports its effect on inhibiting some furin-mediated pathways, but it also highlights that this effect is not caused by direct inhibition of furin [29]. 

In this study, we aimed to discover new compounds that could enlarge the furin inhibitor arsenal. Taking this objective into account, QSAR-based Virtual Screening (QBVS) was carried out to propose potential drugs that could interfere with or block the active center of furin. In addition, this study tested for the first time LMWH’s capacity to directly inhibit furin activity. As a secondary objective, we assayed coadjuvants among compounds from QBVS, LMWH, and CMK as furin inhibitors, since CMK, by inhibition of this enzyme, avoids SARS-CoV-2 S protein cleavage [27]. 

## 2. Results

### 2.1. Virtual Screening

Following the procedure outlined in the methods section, several individual models were developed. Table 1 shows only the best (based on ROCED values) and the most diverse models. The low values of CCR and AUC, and the high values in ROCED of the Y-randomization test show a low-chance correlation in the five models (see Table 2).

With these obtained QSAR models, consensus model virtual screening was carried out against the compounds of natural origin from the FoodBank database (FDB, https://foodb.ca/, accessed on 1 July 2020) to extract possible candidates as furin inhibitors. Table 3 shows the seven most active compounds.

### 2.2. Inhibitors of Furin

Among the compounds highlighted by virtual screening, just Kukoamine A (Cat#CFN93215; ChemFaces, Wuhan, Hubei, China) and Zeaxanthin (Cat#Q444; AK Scientific, Inc., Union City, CA, USA) inhibited both recombinant and physiological furin, and they did so in a dose-dependent manner. In the case of Kukoamine A, the IC50 for recombinant furin was 1.07 mM (95% CI; 0.817–4.577), whereas the IC50 for physiological furin was 193.2 µM (95% CI; 165.5–229) (Figure 1A). With respect to Zeaxanthin, the IC50 for recombinant furin was 90.55 µM (95% CI; 64.87–753), whereas the IC50 for physiological furin was 49.55 µM (95% CI; 38.85–64.52) (Figure 1B). In addition to the compounds selected by virtual screening, furin inhibition by Clexane (LMWH) was also tested. The IC50 for recombinant furin was 31.648 µM (95% CI; 10.08–53.22), whereas the IC50 for physiological furin was 6.45 µM (95% CI; 6.142–6.784) (Figure 1C). 

### 2.3. Adjuvant Effect of Selected Compounds in the Inhibition of Furin

In addition to the effect of the single compounds on the inhibition of furin, we decided to test their coadjuvants with CMK, a known furin inhibitor that also avoids SARS-CoV-2 S protein cleavage [27]. The addition of Zeaxanthin, Kukoamine A, or Clexane IC50 to CMK increased physiological furin inhibition (Figure 2). The IC50 for CMK alone was 4.205 µM (95% CI; 2.991–6.002), whereas the addition of Zexanthin, Kukoamine A or Clexane reduced the IC50 to 1.488 µM (95% CI; 0.885–2.501), 0.749 µM (95% CI; 0.552–0.998) or 0.096 µM (95% CI; 0.037–0.160), respectively. This showed an adjuvant effect on physiological furin inhibition using CMK in combination with Zeaxanthin, Kukoamine A, or Clexane.

### 2.4. Cell Viability in Presence of Zeaxanthin and Kukoamine A

As the effects of CMK and heparin have already been tested on human cell lines [27,30], we decided to check if Zeaxanthin and Kukoamine A could alter cell viability at tested concentrations for physiological furin inhibition. Kukoamine A significantly reduced 293T cells’ viability at double IC50 and full IC50 concentrations (Figure 3). However, a half IC50 concentration did not affect cell viability, as the percentage of live cells was similar to cells in the absence of any compound (87.55% ± 1.48% (half of the IC50) vs. 94.25% ± 3.18% (no Kukoamine A)). In the case of Zeaxanthin, a half IC50 concentration caused a slight reduction in cell viability in comparison to cells in the absence of bioactive compounds (81.1% ± 4.1% (half of the IC50) vs 94.25% ± 3.18% (no Zeaxanthin)) (Figure 3). The Zeaxanthin IC50 concentration reduced cell viability by 61%, whereas double the IC50 concentration reduced it to nearly total death.

### 2.5. Molecular Docking

Furin inhibitors that were predicted by virtual screening and that were experimentally confirmed were submitted to molecular docking calculations, and the resulting poses can be observed in Figure 4 and Figure 5 for Zeaxanthin and Kukoamine A, respectively. It can be appreciated that both compounds extensively blocked access to the active site by means of hydrogen bonds and hydrophobic interactions, thus inhibiting furin’s catalytic action.

## 3. Discussion

Proprotein convertase furin is a very versatile protease. It exhibits different functions that become essential in both health and disease. Among the pathophysiological processes in which furin has a crucial role, there are many types of cancer where furin cleaves and activates a wide range of proteins that promote tumor phenotypes [31]. However, due to its heterogeneous functions, furin represses the tumorigenic properties of some cancer cells and its inhibition can lead to aggressive phenotypes in other tumors [31]. Furin is also involved in some infectious diseases. Viral substrates have gained importance among furin targets as they become activated as infectious proteins when they are cleaved [1]. This relates to viruses from different families, involving infections provoked by Ebola [32], Influenza A [33], or Metapneumovirus [34]. 

In an endeavor to propose furin inhibition as a possible therapeutic target, there have been projects aimed at studying the biological effects of this inhibition in different pathological models. For example, furin autoinhibitory propeptide has been used to reduce metalloproteinase-9 activity in vitro in breast cancer [35,36]. In clinical trials concerning patients with Ewing’s sarcoma cancer, one of the strategies consisted of silencing furin by short hairpin RNAs [37]. In a murine model infected with *Pseudomonas aeruginosa*, nona-D-arginine-mediated furin inhibition diminished corneal adverse effects [38]. In the recent pandemic context, furin inhibition has also been tested as a possible preventive strategy against infection with SARS-CoV-2, by the usage of drugs such as CMK [27]. 

In this study, we aimed to identify new furin inhibitors that could be added to the arsenal of compounds that target furin from a therapeutic point of view. From a virtual screening, we selected a series of candidate compounds, which were tested for the inhibition of purified recombinant and physiological secreted furin. Among these compounds, we selected Zeaxanthin and Kukoamine A, since they were the most effective inhibitors of both physiological and recombinant furin. In addition to these compounds, Clexane, a LMWH, directly inhibited purified recombinant furin but also furin substrate proteolysis in the U-251 MG secretome. These results support that Clexane can directly act against furin. Its IC50 was lower than those of Kukoamine A or Zeaxanthin. Zeaxanthin, as with other carotenoids, is highlighted for its antioxidant and preventive properties against cardiovascular or ocular diseases as well as cancer [39]. In the context of viral infections, lower levels of Zeaxanthin have been reported in the serum of patients infected by the human immunodeficiency virus (HIV), in comparison with noninfected subjects [40]. Although its IC50 value reduced the viability of healthy cells tested in our assay, Zeaxanthin is an FDA-approved drug for which concentrations of up to 100 µM have been administered daily in clinical trials in patients with age-related macular degeneration, with no adverse effects reported [41,42]. In the case of Kukoamine A, some studies support its antitumor properties in glioblastoma [43] and its neuroprotective effects [44]. Similarly, Zeaxanthin’s and Kukoamine A’s IC50 values considerably reduced 293T HEK cell viability. However, some assays that administered Kukoamine A to rats or mice in doses between 5 and 20 mg/kg highlighted only its protective effects, with no significant reported adverse effects [45,46]. Clexane is a well-known venous thromboembolism prophylactic drug [47], but despite its reported abilities to inhibit some furin-mediated pathways, a direct effect of LMWH on this enzyme has not been described [29]. However, our results with purified recombinant furin and Clexane could tip the scale towards a direct interaction. It is important to note that, although we have only tested Clexane, similar effects would be expected with other LMWHs. 

Secondarily, we showed that Kukoamine A, Zeaxanthin, and Clexane increased CMK efficiency over furin, and CMK avoided SARS-CoV-2 S protein proteolysis through furin inhibition [27]. These results suggest that these new inhibitors could be considered, individually or in combination, as potential drugs against COVID-19. In this context, these compounds would bypass the immune escape carried out by SARS-CoV-2 variants, since they act on furin and not on the virus. However, future and extensive studies should be undertaken in order to examine these potential applications. In any case, drug coadjuvants are of vital importance, especially when dealing with infectious diseases, since excessive use of single-compound-based therapies is potentially responsible for the drug-resistant generation of many types of pathogens [48,49,50]. 

The main limitations of this study are related to the high IC50s for the furin inhibitors proposed by virtual screening, taking into account that the QSAR models used to select the compounds were built considering active compounds with IC50/Ki values lower than 0.1 uM. However, there are clinical trials of our selected compounds using concentrations similar to those tested in our experiments. On the other hand, although the selected compounds could potentially be used against COVID-19, it would be necessary to test in clinical trials whether they could prevent contagion and whether they would also be effective in reducing viral replication once contagion has occurred. In addition, we do not know the adverse effects of these drugs in this context. In the specific case of heparin, there are numerous adverse effects described in addition to its contraindication with the use of other drugs [51].

In conclusion, we have identified Kukoamine A, a natural origin compound; Zeaxanthin, an FDA-approved drug; and Clexane, a known antithrombotic compound, as new furin direct inhibitors. In addition, Clexane, Zeaxanthin, and Kukoamine A are able to increase furin inhibition by CMK, so they could also increase its efficiency when avoiding S protein proteolysis and could be tested in clinical trials for COVID-19 prevention.

## 4. Materials and Methods

### 4.1. Virtual Screening

The chemical substances, as well as the activity values (Ki, IC50, and % inhibition), were extracted from the CHEMBL database (https://www.ebi.ac.uk/chembl, accessed on 1 July 2020), for a total of 148 chemicals. All data, both chemicals and activities, were curated for the three activity sets. The data curation was carried out to reduce misannotation and rounding unit errors [52]. The chemical dataset was curated following the protocols proposed by Fourches et al. [53,54]. These protocols included the structural normalization of specific chemotypes, such as aromatic and nitro groups, and the removal of inorganic salts, organometallic compounds, etc. Standardizer was used for chemical curation (Standardizer 17.21.0, ChemAxon https://www.chemaxon.com, Budapest, Hungary, accessed on 1 July 2020). To select the endpoint, the following method was carried out: we used data with values of Ki as the main dataset, using a threshold of pKi ≥ 7 to select actives (45 actives and 66 inactives). Next, we included chemicals with pIC50 values that were not in the main dataset, choosing a threshold of pIC50 ≥ 7 to select active compounds (3 actives and 10 inactives were included). Then, we included chemicals that were not present in the main dataset using % inhibition ≥ 50% as the active selection threshold (0 actives and 7 inactives were included).

More than 5000 different descriptors were calculated for the 131 compounds using DRAGON software [55]. The descriptors with low variance or that were highly correlated with each other (r^2^ > 0.95) were removed. As the QSAR models were based on the principle of similarity (similar substances present similar activities), and to avoid heterogeneities in the data, the so-called activity cliffs (similar substances with very different activity) and outliers (substances very different from the rest of the compounds) were removed. The activity cliff elimination of inactive chemicals was conducted using the Castillo-González et al. [56] method with a threshold value of 101.21. The outlier selection was carried out with the applicability domain (AD) protocol defined by Melagraki et al. [57,58] with Z = 0.5 (see applicability domain section). Once the analysis was completed, 18 compounds were eliminated (15 activity cliffs and 3 outliers). To ensure that the data will result in effective QSAR modeling, they must present a MODI value [59] greater than 0.65; for our data, the value obtained from MODI was 0.76.

The 113 chemicals selected were randomly divided into three subsets: training (80%), test (10%), and external (10%). With these three subsets, we used linear discriminant analysis for the QSAR model development and a genetic algorithm for the descriptor selection technique using the ROCED [60] parameter as a fitness function. We computed models using between 2 and 11 descriptors. Moreover, selectivity, specificity, correct classification rate (CCR), and area under the ROC curve (AUC) were calculated for training, test, and external sets. 

In addition, the leave-one-out (LOO) approach was conducted on the training set to assess the internal predictivity. In this approach, one compound of the training set was omitted, and the statistical parameters were recalculated with the remaining substances. This process was conducted for all training sets, and the values of the sensitivity, specificity, AUC, and ROCED were reported. High values of sensitivity and specificity and an AUC with a low value for ROCED are indicative of a model’s robustness.

To avoid models producing good classification due to chance, the Y-randomization technique was carried out. The activity values of the training set were randomized (Yrnd). This process was conducted 300 times and the average values of the sensitivity (Sensrnd), specificity (Sprnd), and ROCEDrnd for both training and test sets, as well as the AUC values, were reported. A high average in ROCED and a low average in sensitivity, specificity, and AUC are indicative of a good QSAR model. 

To obtain the most statistically robust and predictive models, we employed the combinatorial QSAR strategy. For this purpose, the most different models based on the canonical measure of distance (CMD) [61] were chosen to preserve the most information and diversity and to construct a consensus model. 

In this study, we defined AD as a distance threshold ΔT between a compound of interest and its nearest neighbors of the set considered. It was calculated as
(1)$$\Delta T=\bar{y}Z\sigma$$
where $\bar{y}$ and σ are the mean and the standard deviation of the distances that are below the mean of the distance matrix, and Z is defined at 0.5. For outlier exclusion, we defined the AD in the entire descriptor space, and a chemical that presented a Euclidean distance with its nearest neighbor that was higher than this value was considered as an outlier. For chemical prediction in the virtual screening test, we defined the AD in the selected descriptor space of the model, and for chemicals that presented a Euclidean distance with their nearest neighbors in the training set higher than this value, no prediction was made. Compound class (i.e., active or inactive) assignment in the virtual screening was based on the majority vote across the independent models developed with one condition: the chemical must be inside the applicability domain defined above. To prioritize compounds with the same activity vote, we sorted them by the mean value of probability given by each model. 

### 4.2. Secretome Collection from U-251 MG Cells

The human glioblastoma cell line U-251 MG expresses human physiological furin, which can be secreted or retained intracellularly [62]. The U-251 MG cells (European Collection for Authenticated Cell Cultures) were first cultured in Corning^®^ T-75 flasks (catalog #430641; ThermoFisher Scientific, Waltham, MA, USA) with DMEM (Dulbecco’s Modified Eagle’s Medium) containing 4.5 g/L glucose (Gibco Thermo Fisher, Madrid, Spain), supplemented with 10% fetal bovine serum (Gibco Thermo Fisher, Madrid, Spain), 1% GlutaMax (Gibco Thermo Fisher, Madrid, Spain), 1% nonessential amino acids (Gibco Thermo Fisher, Madrid, Spain) and 0.1% gentamicin (Gibco Thermo Fisher, Madrid, Spain). When 100% confluence was reached, the medium was recovered and centrifuged for 5′ at 280 g to discard cell debris, and finally, supernatant (secretome) was collected. The activity of secreted furin was confirmed by using a fluorogenic specific substrate (pERTKR-AMC fluorogenic peptide substrate; ES013; LOT#PYO02; R&D Systems, Minneapolis, MN, USA) (Figure A1). These results were achieved by incubating U-251 MG secretome concentrated 10X with 50 µM furin substrate in a 96-well black plate (10030581, ThermoFisher Scientific, Waltham, MA, USA), and then, measuring fluorescence in a Multiskan Go (ThermoFisher Scientific, Waltham, MA, USA) plate reader at 380 and 460 nm for emission and excitation wavelengths, respectively, at 37 °C for 30 min.

### 4.3. Furin Inhibition Assay

Clexane (LMWH, enoxaparin sodium, Clexane, Sanofi Aventis S.A., Barcelona, Spain) and each candidate compound selected by virtual screening were assayed as inhibitors of human recombinant furin (1503-SE-010; Lot#INK2320031; R&D Systems) and physiological furin secreted from U-251 MG cells. For both types of furin, the fluorogenic specific substrate (mentioned above) was used. In the inhibition assays, activity buffer (Tris Base 25 mM (BP152-1; LOT#165920; Fisher BioReagents), CaCl2 1 mM and Brij-35 (Cat. No. 20150; ThermoFisher Scientific, Waltham, MA, USA) 0,5% (*w/v*), pH 9) was the medium for recombinant furin, whereas distilled water was used for U-251 MG secretome since it showed less furin activity in the presence of activity buffer. Compounds were reconstituted in their appropriate solvents (distilled water, ethanol, PBS, etc.), according to the supplier’s instructions. In a 96-well black plate, U-251 MG secretome concentrated 10× or 3.6 nM human recombinant furin were incubated with different compound concentrations for 10 min at room temperature, and then, 50 µM furin specific fluorogenic substrate was added to every well, except one blank. Fluorescence was recorded in the Multiskan Go plate reader at 380 and 460 nm for emission and excitation wavelengths, respectively, at 37 °C for 30 min.

### 4.4. IC50 Calculation

The IC50 was calculated only for those compounds that inhibited recombinant and physiological furin. For each compound concentration, furin activity’s maximum velocity (mRFU/min) was recorded. Then, by using GraphPad software (v. 8), the correlation between the decimal logarithm (compound concentration) and furin activity’s maximum velocity was calculated by nonlinear regression (curve fit) for XY analyses. From this statistical method, the IC50 was calculated with a 95% confidence interval (CI).

### 4.5. Furin Inhibition Assay in Coadjuvants 

Since furin inhibition by CMK had been tested only in physiological conditions [27], adjuvant inhibition assays were performed with furin from the U-251 MG secretome. Inhibition was assessed by using different concentrations of CMK (Cat.No. B5437; APExBIO) and the IC50 concentrations of selected virtual screening compounds or Clexane. In a 96-well black plate, various CMK (0–50 µM) concentrations were incubated with the IC50 concentrations of the rest of the compounds and with 10× U-251 MG secretome as described above. After 10 min incubation at room temperature, 50 µM furin specific fluorogenic substrate was added and the plate was read in a Multiskan Go plate reader at 380 and 460 nm for emission and excitation wavelengths, respectively, at 37 °C for 30 min.

### 4.6. Cell Viability in Presence of Selected Furin Inhibitors

New furin inhibitors identified by virtual screening were tested in cell viability assays. The 293T HEK cell line (human embryonic kidney cell line purchased from American Type Culture Collection) was used as healthy cell model. They were cultured in DMEM containing 4,5 g/L glucose, supplemented with 10% fetal bovine serum, 1% GlutaMax, and 0,1% gentamicin. They were maintained in Corning^®^ T-75 flasks (catalog #430641; ThermoFisher Scientific, Waltham, MA, USA) at 37 °C and 5% carbon dioxide. At 100% cellular confluence, cells were subcultured in 12-well plates (flat bottom, sterile, NUNC brand, Biolab, Madrid, Spain) at 100,000 cells/mL for 24 h. Then, different concentrations (double, half, and full IC50 concentrations for physiological furin) of furin inhibitors were added to cells for 24 h, and cells were collected afterward. The experiment was run in duplicates. Then, the cell pellet was resuspended in running buffer (autoMACS ^®^ Running Buffer–MACS Separation Buffer; order no.:130-091-221; Miltenyi Biotec, Madrid, Spain) and 7-AAD (Cat: 51-68981E; BD Biosciences, Madrid, Spain). Finally, this solution was evaluated in a flow cytometer (BD Accuri™ C6 Plus Flow Cytometer; BD Biosciences, Madrid, Spain) to calculate mortality rates. A blank solution with cell pellet resuspended in running buffer was used to discard cell autofluorescence. The results were analyzed in FlowJo software (v. 10).

### 4.7. Molecular Docking

To provide mechanistic insights into the structure and main interactions established between furin and characterized inhibitors derived from virtual screening, molecular docking calculations were carried out.

The selected inhibitors were set up for docking simulations using AmberTools (AMBER 2017, University of California, San Francisco, CA, USA) [63]. Molecular parameters were calculated by computing partial charges by the MMFF94 force field, by adding hydrogen atoms, and by minimizing energies (default parameters) [64]. 

The crystal structure of furin (Protein Data Bank code 5MIM) was used to build the protein model system. At an early stage, bond orders were assigned, hydrogens were added, and cap termini were included with the Protein Preparation Wizard module as implemented in Maestro (Schrödinger Release 2021-2: Maestro, Schrödinger, LLC, New York, NY, USA) [65]. Protonation states of all side chains were subsequently defined using PROPKA3.1. Partial charges over all atoms were finally assigned within the AMBER99 force field scheme as implemented in AmberTools. Docking simulations were performed with Lead Finder software v1.1.20 [66] via MetaScreener (https://github.com/bio-hpc/metascreener, accessed on 20 December 2021) in the coordinates of the active site of the protein. All docking parameters were set to default for the calculations. The best-ranked docking score pose for every compound was retained for further analysis.

## 5. Patents

Two patents in connection with these findings were filed in Spain, 20 September 2021: (a) 10. H. Pérez-Sánchez, I. Martínez-Martínez, D. Zaragoza-Huesca, C. Martínez-Cortés, A.J. Banegas-Luna, A. Pérez-Garrido, J.M. Vegara-Meseguer, J. Peñas-Martínez, M. C. Ródenas-Bleda, S. Espín-García, “Zeaxantina para la prevención y tratamiento de la infección viral, preferiblemente por coronavirus”, P202130873 (2021), and (b) 11. H. Pérez-Sánchez, I. Martínez-Martínez, D. Zaragoza-Huesca, C. Martínez-Cortés, A.J. Banegas-Luna, A. Pérez-Garrido, J. Vegara-Meseguer, J. Peñas-Martínez, M. C. Ródenas-Bleda, S. Espín-García, “Kukoamina A para la prevención y tratamiento de la infección viral, preferiblemente por coronavirus”, P202130872 (2021).

## Figures and Tables

**Figure 1 ijms-23-02796-f001:**
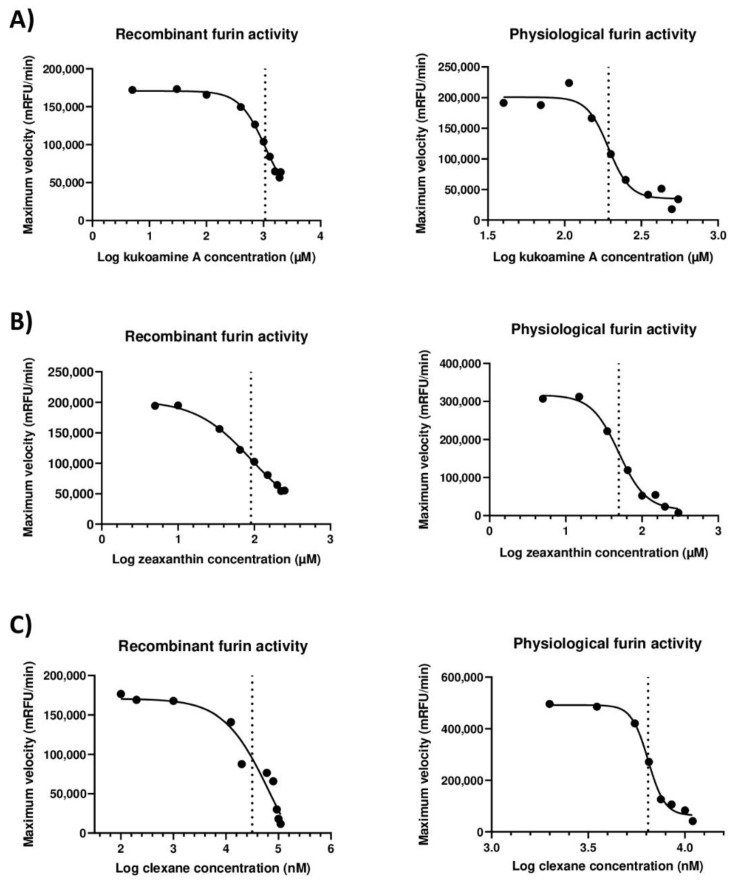
Dose-dependent inhibition of furin activity by Kukoamine A, Zeaxanthin, and Clexane. (**A**) Kukoamine A inhibited the activity of both recombinant and physiological furin. (**B**) Zeaxanthin inhibited the activity of both recombinant and physiological furin. (**C**) Clexane inhibited the activity of both recombinant and physiological furin. Vertical dotted line represents IC50 value for each compound. *mRFU/min*: millirelative fluorescence units/minute; *Log:* decimal logarithm.

**Figure 2 ijms-23-02796-f002:**
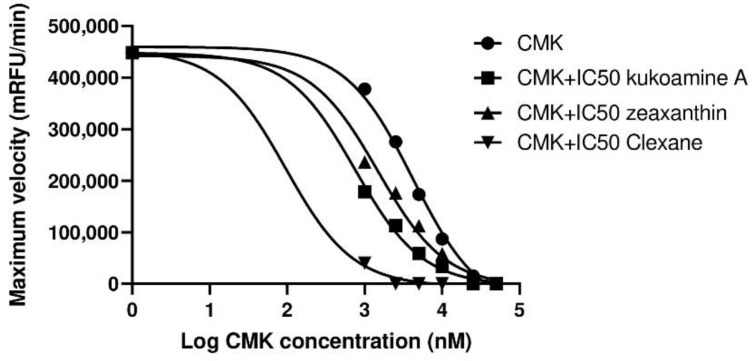
Adjuvant inhibition of physiological furin using CMK in combination with Clexane, Kukoamine A, or Zeaxanthin. *mRFU/min*: millirelative fluorescence units/minute; *Log*: decimal logarithm.

**Figure 3 ijms-23-02796-f003:**
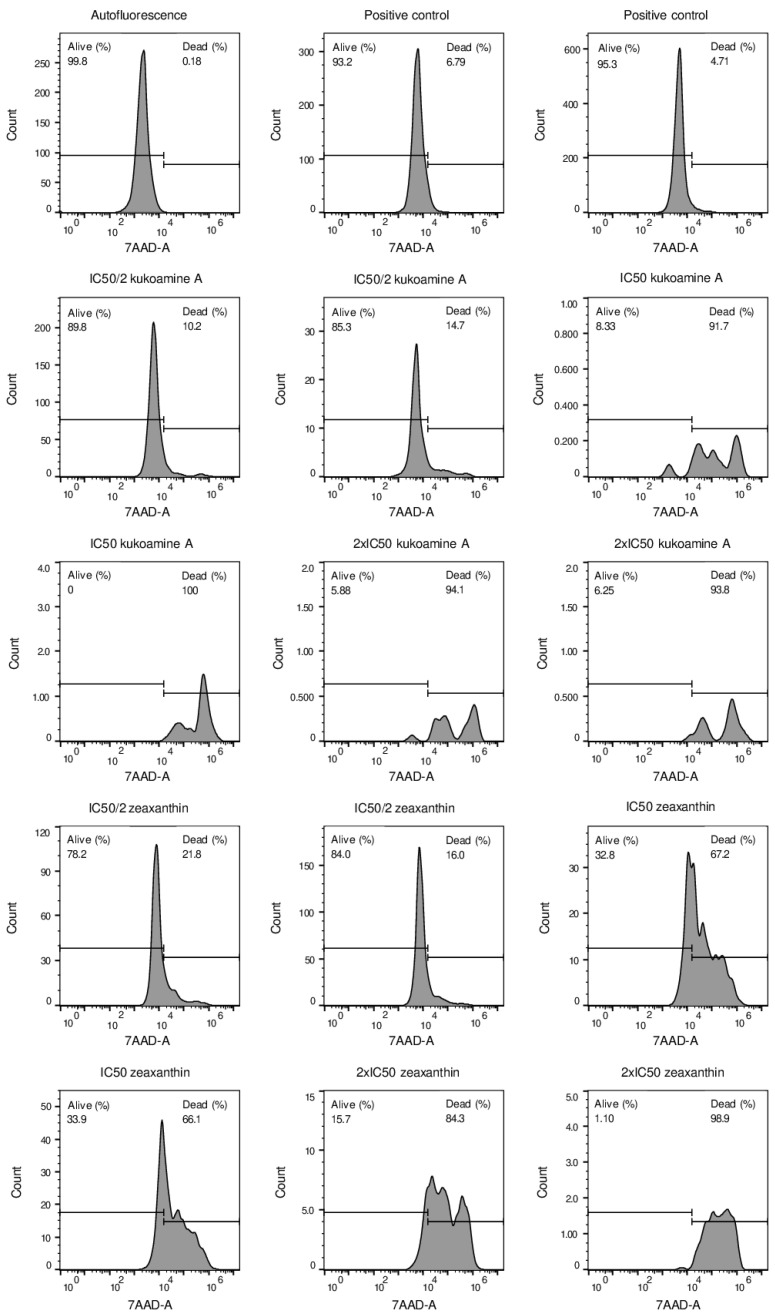
Viability percentage of 293T HEK cells at different Kukoamine A or Zeaxanthin concentrations. Y-axis represents cell count by cytometer, whereas X-axis shows 7-AAD fluorescence intensity. Autofluorescence sample delimits 7-AAD positive and negative cell regions. Positive control represents cells without bioactive compounds. Each sample type was duplicated.

**Figure 4 ijms-23-02796-f004:**
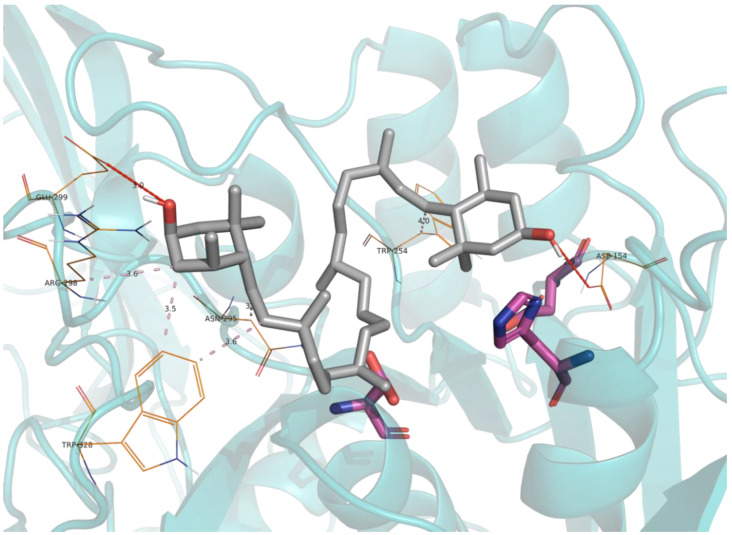
Obtained docking pose for Zeaxanthin into the active site of furin. The carbon skeleton of Zeaxanthin is depicted in grey, while the carbon skeleton of catalytic triad residues is depicted in pink. Main interaction residues are shown in lines represented in orange. Hydrogen interactions are represented by red dashed lines, while hydrophobic interactions are shown by purple dashed lines. The secondary structure of the protein is shown in cartoon representation.

**Figure 5 ijms-23-02796-f005:**
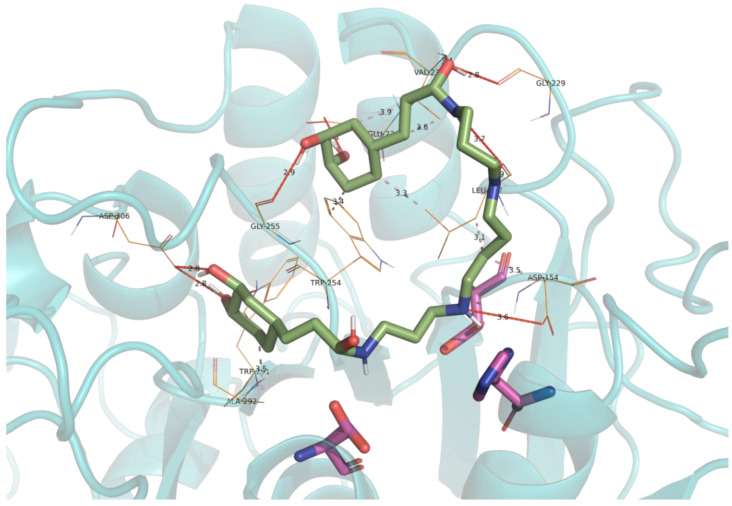
Obtained docking pose for Kukoamine A into the active site of furin. The carbon skeleton of Kukoamine A is depicted in green, while the carbon skeleton of catalytic triad residues is depicted in pink. Main interaction residues are shown in lines represented in orange. Hydrogen interactions are represented by red dashed lines, while hydrophobic interactions are shown by purple dashed lines. The secondary structure of the protein is shown in cartoon representation.

**Table 1 ijms-23-02796-t001:** Best QSAR models selected and consensus model with its statistical parameters (only values of CCR, AUC, and ROCED are shown).

		Train		Test			External		
	n	CCR ^a^	AUC ^b^	CCR	AUC	ROCED	CCR	AUC	ROCED
*QSAR1*	6	0.96	0.97	0.92	0.89	0.27	0.73	0.82	0.81
*QSAR2*	6	0.96	0.96	0.92	0.9	0.27	0.79	0.8	0.66
*QSAR3*	8	0.91	0.96	0.92	0.86	0.34	0.7	0.77	1.04
*QSAR4*	9	0.98	0.97	0.92	0.85	0.25	0.73	0.71	0.79
*QSAR5*	11	0.98	0.98	0.9	0.86	0.23	0.7	0.7	0.91
*Consensus*	-	0.98	1	0.89	1	0.35	0.73	1	0.79

^a^ CCR: correct classification rate; ^b^ AUC: area under ROC curve.

**Table 2 ijms-23-02796-t002:** Leave-one-out cross-validation and Y-randomization parameters of QSAR models.

	LOO_cv_ ^a^		Train		Test			External		
	CCR ^b^	ROCED	CCR	AUC ^c^	CCR	AUC	ROCED	CCR	AUC	ROCED
*QSAR1*	0.93	0.19	0.62	0.67	0.5	0.59	2.67	0.5	0.65	2.72
*QSAR2*	0.89	0.31	0.63	0.68	0.5	0.59	2.68	0.5	0.65	2.89
*QSAR3*	0.79	0.67	0.65	0.71	0.5	0.59	2.68	0.5	0.61	2.71
*QSAR4*	0.89	0.24	0.66	0.72	0.5	0.6	2.69	0.5	0.62	2.77
*QSAR5*	0.86	0.35	0.68	0.75	0.5	0.61	2.72	0.49	0.6	2.76

^a^ LOO_cv_: leave-one-out cross validation; ^b^ CCR: correct classification rate; ^c^ AUC: area under ROC curve.

**Table 3 ijms-23-02796-t003:** List of compounds obtained in the consensus model virtual screening.

FDB ID ^a^	Compound	Activity Vote ^b^	Probability Mean ^c^	(%) in Domain ^d^
FDB030264	15,15′-dihydroxy-β-carotene	1	0.897	100
FDB001534	Lactucaxanthin	1	0.867	100
FDB002245	Kukoamine A	1	0.867	100
FDB002479	(3S,3′R,4xi)-beta,beta-Carotene-3,3′,4-triol	1	0.865	100
FDB007276	Lutein ester	1	0.865	100
FDB014726	Zeaxanthin	1	0.863	100
FDB015828	7,8-Dehydroastaxanthianthin	1	0.862	100

^a^ The FDB identification number; ^b^ the activity classification of the consensus model; ^c^ the mean of probabilities for the five models selected (consensus model); ^d^ the percentage of five models (consensus model) in which the compounds are in the applicability domain.

## Data Availability

The original data are available upon request to the authors.

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
