# Peer review of "Identification of Kukoamine A, Zeaxanthin, and Clexane as New Furin Inhibitors"

_ijms, 2022, doi:10.3390/ijms23052796_

Round 1
Reviewer 1 Report
The manuscript by Zaragoza-Huesca et al. “Identification of Kukoamine A, Zeaxanthin and Clexane as furin inhibitors and potential preventive drugs for the infection by SARS-CoV-2” is noteworthy. It requires minor revision for publication in the International Journal of Molecular Sciences.
Comments.
- Lines 54-65, please add informations on COVID-19 infections, mortality, prevention strategies as role of health and immunity and biomolecules i.e. doi: 10.1007/s12088-020-00908-0 and doi: 10.1007/s12088-020-00893-4.
- Introduction, please highlight the challenges about COVID-19 variants and also discuss them i.e. https://doi.org/10.1007/s15010-021-01734-2.
- Figs quality can be improved.
Author Response
The manuscript by Zaragoza-Huesca et al. “Identification of Kukoamine A, Zeaxanthin and Clexane as furin inhibitors and potential preventive drugs for the infection by SARS-CoV-2” is noteworthy. It requires minor revision for publication in the International Journal of Molecular Sciences.
We would like to thank the reviewer for his/her kind words and for giving us the opportunity to improve our research with his/her suggestions.
Comments.
- Lines 54-65, please add informations on COVID-19 infections, mortality, prevention strategies as role of health and immunity and biomolecules i.e. doi: 10.1007/s12088-020-00908-0 and doi: 10.1007/s12088-020-00893-4.
As suggested by the reviewer, this information has been included as well as the reference (Page2).
- Introduction, please highlight the challenges about COVID-19 variants and also discuss them i.e. https://doi.org/10.1007/s15010-021-01734-2.
The information suggested by the reviewer has been included (Page 2 and 10).
- Figs quality can be improved.
Quality of figures has been improved as suggested
All changes have been highlighted in yellow for easy detection
Reviewer 2 Report
In the present manuscript, the authors identified novel compounds that could inhibit the catalytic activity of furin, which is considered an enzyme implicated in SARS-CoV-2 infections. Overall, the study is well designed and supported by valid and reasoned conclusions. The manuscript is well written and addresses a topic that covers not only the pandemic COVID but also other pathologies involving furin.
Here are my comments.
“In addition, since its emergence, more than 508000 deaths were detected in nearly 200 different regions [22].” The weekly epidemiological update on COVID-19 (25 January 2022) from WHO reports over 5.5 million deaths worldwide (https://www.who.int/publications/m/item/weekly-epidemiological-update-on-covid-19---25-january-2022). Please update.
In the Introduction the authors present drugs that have been studied as potential furin inhibitors. Diminazene, Decanoyl-Arg-Val-Lys-Arg-chloromethylketone (CMK), modified α1-antitrypsin Portland, and low molecular weight heparin (LMWH) are mentioned, but only the discoveries on CMK and LMWH are presented. What about diminazene and modified α1-antitrypsin Portland? What was their effect and mechanism on furin?
Why has CMK not been tested for inhibition of both recombinant and physiological furin? It would be a good control because it was discovered to inhibit furin from cleaving SARS-CoV-2? I really miss this comparison and think it would be very important for evaluation of compounds.
Line 253: “Main limitations of this study are related to high IC50 for furin inhibitors proposed by virtual screening,…” It seems that I am missing some information. Where are this IC50 stated?
Line 280: “We used the data with values of Ki as a main dataset using a threshold of pKi >= 7 to select actives (45 actives and 66 inactives). Following, we included chemical with IC50 values that there were not in the main dataset choosing a threshold of IC50>=7 to select active compounds (3 actives and 10 inactives were included). Then, we included chemicals that were not present in the main dataset using % inhibition ≥ 50% as active selection threshold (0 actives, and 7 inactives were included).” I think the authors meant pIC50 = > 7 and not IC50. What was the concentration at which the compounds showed furin inhibition ≥ 50%? Since the thresholds for Ki and IC50 were chosen based on the same concentration, the concentration at which inhibition was measured should also be the same.

Author Response
In the present manuscript, the authors identified novel compounds that could inhibit the catalytic activity of furin, which is considered an enzyme implicated in SARS-CoV-2 infections. Overall, the study is well designed and supported by valid and reasoned conclusions. The manuscript is well written and addresses a topic that covers not only the pandemic COVID but also other pathologies involving furin.
Here are my comments.
We would like to thank the reviewer for his/her kind words and for giving us the opportunity to improve our research with his/her suggestions.
“In addition, since its emergence, more than 508000 deaths were detected in nearly 200 different regions [22].” The weekly epidemiological update on COVID-19 (25 January 2022) from WHO reports over 5.5 million deaths worldwide (https://www.who.int/publications/m/item/weekly-epidemiological-update-on-covid-19---25-january-2022). Please update.
As suggested by the reviewer, we have updated this information (Page 2).
In the Introduction the authors present drugs that have been studied as potential furin inhibitors. Diminazene, Decanoyl-Arg-Val-Lys-Arg-chloromethylketone (CMK), modified α1-antitrypsin Portland, and low molecular weight heparin (LMWH) are mentioned, but only the discoveries on CMK and LMWH are presented. What about diminazene and modified α1-antitrypsin Portland? What was their effect and mechanism on furin?
We have included some information in the Introduction to describe the inhibitory mechanism of diminazene and modified α1-antitrypsin Portland as suggested.
Why has CMK not been tested for inhibition of both recombinant and physiological furin? It would be a good control because it was discovered to inhibit furin from cleaving SARS-CoV-2? I really miss this comparison and think it would be very important for evaluation of compounds.
Inhibition of recombinant furin was assayed in order to check whether compounds identified by virtual screening, and also clexane, were able to directly inhibit furin activity. In the case of CMK, it has already demonstrated that it directly inhibits furin. In addition, authors demonstrated its potential as inhibitor of SARS-CoV-2 spike cleavage by using a cellular model which expressed endogenous furin (Cheng, Y.W.; et al. Cell Rep 2020, 33(2), 108254. DOI: 10.1016/j.celrep.2020.108254). We additionally tested coadjuvants between CMK and the discovered compounds in the inhibition of cell endogenous furin to check the potential application as furin inhibitors of SARS-CoV-2 spike protein cleavage.
Line 253: “Main limitations of this study are related to high IC50 for furin inhibitors proposed by virtual screening,…” It seems that I am missing some information. Where are this IC50 stated?
This information is included in page 4 (section 2.2 Inhibitors of furin).
Line 280: “We used the data with values of Ki as a main dataset using a threshold of pKi >= 7 to select actives (45 actives and 66 inactives). Following, we included chemical with IC50 values that there were not in the main dataset choosing a threshold of IC50>=7 to select active compounds (3 actives and 10 inactives were included). Then, we included chemicals that were not present in the main dataset using % inhibition ≥ 50% as active selection threshold (0 actives, and 7 inactives were included).” I think the authors meant pIC50 = > 7 and not IC50. What was the concentration at which the compounds showed furin inhibition ≥ 50%? Since the thresholds for Ki and IC50 were chosen based on the same concentration, the concentration at which inhibition was measured should also be the same.
This is a typo, we were referring to pIC50 and not IC50. We have corrected them in the manuscript. The percentage of inhibition was mainly used to include more negative substances to balance the dataset (see the number of chemicals within class included in the main dataset), since the concentration that is usually used in inhibition assays is higher than used in Ki and IC50.
All changes have been highlighted in yellow for easy detection
Reviewer 3 Report
The manuscript entitled "Identification of Kukoamine A, Zeaxanthin and Clexane as new furin inhibitors" describes the results of selecting potential furin inhibitors by molecular modeling (QSAR) methods, then the evaluation of the three compounds, and the results of the adjuvant effect of these compounds on the known furin inhibitor decanoyl-RVKR- co may have a positive effect on SARS-CoV-2 therapy.
The results are interesting, I propose a manuscript correction.
Table 3.
The first column shows the FDB identification number of the compound. The name of the compound shall be added or the name and chemical structure of the selected potential furin inhibitors shall be included in the supporting information.
Three selected compounds (Kukoamine A, Zeaxanthin and Clexane) showed furin inhibitory activity - both in recombinant and physiological form, however the activity of the tested compounds was different.
Compared to the activity of decanoyl-RVKR- (CMK) with an IC50 value of 4.205 micromol, the potency of the test compounds was lower, with an IC50 value ranging from 293.2 to 49.55 micromol.
Author Response
The manuscript entitled "Identification of Kukoamine A, Zeaxanthin and Clexane as new furin inhibitors" describes the results of selecting potential furin inhibitors by molecular modeling (QSAR) methods, then the evaluation of the three compounds, and the results of the adjuvant effect of these compounds on the known furin inhibitor decanoyl-RVKR- co may have a positive effect on SARS-CoV-2 therapy.
The results are interesting, I propose a manuscript correction.
We really appreciate the kind comments of the reviewer on our manuscript. We also acknowledge his/her suggestions and corrections.
Table 3.
The first column shows the FDB identification number of the compound. The name of the compound shall be added or the name and chemical structure of the selected potential furin inhibitors shall be included in the supporting information.
As suggested by the reviewer the name of the compound has been included in the table as well as the link to the website where the chemical structure is included.
Three selected compounds (Kukoamine A, Zeaxanthin and Clexane) showed furin inhibitory activity - both in recombinant and physiological form, however the activity of the tested compounds was different.
The reviewer is right. One possible explanation consists on that specific cell post-translational modifications may influence furin activity. In this context, both furin proteins used have been expressed in different tumor cell lines (human glioblastoma for physiological furin and mouse myeloma for recombinant furin). In addition, recombinant furin has a C-terminal 10-His tag that might slightly impair the accession of compounds to the furin active site.
Apart from differences respecting to protein structure, activity buffer used for recombinant furin has a pH=9 whilst physiological furin solved at glioblastoma secretome was assayed at pH=7 as already stated in page 12 (section 4.3 Furin inhibition assay).
Compared to the activity of decanoyl-RVKR- (CMK) with an IC50 value of 4.205 micromol, the potency of the test compounds was lower, with an IC50 value ranging from 293.2 to 49.55 micromol.
We agree IC50 value of Kukoamine A (193.2 micromol) and Zeaxanthin (49.55 micromol) are higher than CMK one (4.205 micromol), although our study also highlights combination of these compounds reduce CMK IC50 over furin.
All changes have been highlighted in yellow for easy detection